# Nutraceuticals and Cancer: Potential for Natural Polyphenols

**DOI:** 10.3390/nu13113834

**Published:** 2021-10-27

**Authors:** Jessica Maiuolo, Micaela Gliozzi, Cristina Carresi, Vincenzo Musolino, Francesca Oppedisano, Federica Scarano, Saverio Nucera, Miriam Scicchitano, Francesca Bosco, Roberta Macri, Stefano Ruga, Antonio Cardamone, Annarita Coppoletta, Annachiara Mollace, Francesco Cognetti, Vincenzo Mollace

**Affiliations:** 1IRC-FSH Department of Health Sciences, University “Magna Græcia” of Catanzaro, 88100 Catanzaro, Italy; jessicamaiuolo@virgilio.it (J.M.); micaela.gliozzi@gmail.com (M.G.); carresi@unicz.it (C.C.); xabaras3@hotmail.com (V.M.); oppedisanof@libero.it (F.O.); federicascar87@gmail.com (F.S.); saverio.nucera@hotmail.it (S.N.); miriam.scicchitano@hotmail.it (M.S.); boscofrancesca.bf@libero.it (F.B.); robertamacri85@gmail.com (R.M.); rugast1@gmail.com (S.R.); tony.c@outlook.it (A.C.); annarita.coppoletta@libero.it (A.C.); 2Nutramed S.c.a.r.l, Complesso Ninì Barbieri, Roccelletta di Borgia, 88021 Catanzaro, Italy; 3Medical Oncology 1, Regina Elena National Cancer Institute, IRCCS, 00144 Rome, Italy; annachiaramollace@gmail.com (A.M.); fcognetti@gmail.com (F.C.); 4IRCCS San Raffaele, Via di Valcannuta 247, 00133 Rome, Italy

**Keywords:** polyphenols, bergamot, oleuropein, quercetin, curcumin, apoptosis, inflammation, antioxidant property

## Abstract

Cancer is one of the leading causes of death globally, associated with multifactorial pathophysiological components. In particular, genetic mutations, infection or inflammation, unhealthy eating habits, exposition to radiation, work stress, and/or intake of toxins have been found to contribute to the development and progression of cancer disease states. Early detection of cancer and proper treatment have been found to enhance the chances of survival and healing, but the side effects of anticancer drugs still produce detrimental responses that counteract the benefits of treatment in terms of hospitalization and survival. Recently, several natural bioactive compounds were found to possess anticancer properties, capable of killing transformed or cancerous cells without being toxic to their normal counterparts. This effect occurs when natural products are associated with conventional treatments, thereby suggesting that nutraceutical supplementation may contribute to successful anticancer therapy. This review aims to discuss the current literature on four natural bioactive extracts mostly characterized by a specific polyphenolic profile. In particular, several activities have been reported to contribute to nutraceutical support in anticancer treatment: (1) inhibition of cell proliferation, (2) antioxidant activity, and (3) anti-inflammatory activity. On the other hand, owing to their attenuation of the toxic effect of current anticancer therapies, natural antioxidants may contribute to improving the compliance of patients undergoing anticancer treatment. Thus, nutraceutical supplementation, along with current anticancer drug treatment, may be considered for better responses and compliance in patients with cancer. It should be noted, however, that when data from studies with bioactive plant preparations are discussed, it is appropriate to ensure that experiments have been conducted in accordance with accepted pharmacological research practices so as not to disclose information that is only partially correct.

## 1. Introduction

To date, it is well-known that cancer is one of the leading causes of death globally. The report entitled Global Cancer Statistics 2020, produced in collaboration with the American Cancer Society (ACS) and the International Agency for Research on Cancer (IARC), confirmed that in 2020 about 17 million people were affected by this disease, comprising 36 types of cancer in 185 countries around the world [1,2]. As a result, the annual global health expenditure is extremely high. For this reason, projections of cancer incidence and mortality are crucial to understanding the evolving scenario of cancer risk. Within a population, the number of individuals who are diagnosed or die of cancer is largely influenced by age, environment, and lifestyle. A very large study considered the incidence and mortality of 26 types of cancer and highlighted their evolution from 1993 to 2014, and made projections from 2014 to 2035. Obviously, the projections do not include assumptions about changes in risk factors [3]. Figure 1 shows 6 of the 26 types of cancer reported in this study.

Cancer is a disease in which some cells grow uncontrollably and can spread to other parts of the body. In fact, it is important to stress that not all tumors are cancerous: benign tumors are characterized by cells that do not show signs of transformation and remain confined to the site of origin. On the contrary, the main characteristic of malignant tumors (cancer) is the ability of the cells that constitute them to migrate from the original site to a secondary location and metastasize to adjacent tissues, organs, and/or different parts of the body through lymphatic or hematogenic diffusion [4]. Cancer has a multifactorial origin, and its causes are found in genetic mutations, infection or inflammation, unhealthy eating habits, exposure to radiation, work stress, and/or intake of toxins [5]. Before achieving the aggressiveness necessary to become life-threatening, a tumor must be able to: (a) replicate limitlessly; (b) move; (c) evade apoptosis; (d) produce growth signals that are self-sufficient; (e) be insensitive to anti-growth signals; (f) degrade the extracellular matrix; (g) survive in the blood; (h) share in the environment of a new tissue [6]. Normal cells can transform into cancerous cells, but before this happens, they must undergo the phenomena of abnormal changes known as hyperplasia and dysplasia. In hyperplasia, there is a considerable increase in the number of cells that maintain normal characteristics. In contrast, cells assume abnormal phenotypic characteristics in dysplasia. It is important, however, to point out that hyperplasia and dysplasia do not necessarily cause cancer [7]. In general, early detection of cancer and proper treatment increase the chances of survival and healing. The type of cancer and the stage suggest the most suitable treatment to use; treatment options may be chemotherapy, surgery, radiotherapy, hormonal therapy, targeted therapy, etc. Today, it is particularly appropriate to use a combination of treatment methods to ensure the maximum effectiveness and optimal results [8]. However, each treatment has its side effects on the patient, and the oncologist should choose the most appropriate treatment, considering the risk–benefit ratio [9]. Chemotherapy is generally accepted as the standard therapy and remains one of the main strategies in the treatment of primary tumors, although it is well-known to cause DNA damage and affect both cancerous and non-cancerous cells. In addition, cardiotoxicity is a complication of this treatment; the severity of cardiotoxicity is dependent on cumulative dose, the type and combination of drugs used, and the presence of co-existing pathologies such as diabetes mellitus, cardiac diseases, and other risk factors [10,11]. Radiation therapy also has side effects, such as neurological deficits caused by vascular damage and fibrosis of neuronal structures [12,13]. Hormone therapy can be used to manage hormone-dependent malignant tumors, to manipulate the endocrine system, and to interfere with hormonal production or the activity of their receptors. In general, hormone therapy involves the administration of exogenous hormones such as corticosteroids, selective estrogen receptor modulators, somatostatin analogs, progestins, gonadotropin-releasing hormone agonists and antagonists, aromatase inhibitors, and antiandrogens. Some of these have antiproliferative and pro-apoptotic effects. Unfortunately, this treatment can cause a wide range of complications including liver steatosis, thrombosis, endometrial and osteoporosis hypertrophy, intestinal perforation, pulmonary embolism, vascular necrosis, and breast and endometrial cancers [14]. Surgical resection is still widely used in cancer treatment as it effectively relieves the patient’s symptoms. However, much scientific evidence has shown that cancer recurs in many patients after a short time, owing to the stress induced by surgery, which at the systemic level, stimulates inflammation, increased release of cytokines, and the risk of cancer recurrence [15]. In addition, surgical resection potentially enhances metastatic seeding of tumor cells, spreading cancer cells in the vascular and lymphatic systems and favoring their migration into distant organs [16].

## 2. Natural Compounds and Cancer

Since the main purpose of anticancer treatments is to kill cancer cells without damaging normal cells, and these drugs exert their action aspecifically on both cancer and normal cells, it is necessary to develop an effective treatment with anticancer properties and minor adverse effects [17]. More and more patients choose a non-traditional anticancer treatment in addition to conventional chemotherapy and radiation therapy; here we discuss the use of a complementary medicine that combines conventional and unconventional approaches. Alternative medicine is the term used when unconventional treatment completely replaces conventional treatment modes, emphasizing the mental, emotional, spiritual, and social aspects of the patient [18]. Risk factors related to cancer onset include not only inheritance, exposure to harmful substances, and hormonal imbalance, but also lifestyle, including diet and nutrition. Dietary schemes based on regular intake of fruit, vegetables, foods rich in selenium, vitamins (B-12 or D), folic acid, and antioxidants, along with high intake of fiber-rich products and moderate consumption of milk and dairy products, play a protective role in the prevention of cancer. On the other hand, consumption of meat and animal products or animal fats may increase the incidence of cancer [19]. In particular, the World Health Organization (WHO) highlights that a balance of energy intake—with a heightened intake of fruit and vegetables and limited consumption of saturated fats, sugar, and salt—greatly reduces the risk factors related to the onset of diseases [20]. So, the concept of the “healthy diet” has been developed, which corresponds to a food plan that is able to guarantee health. In a healthy diet, macronutrients (carbohydrates, proteins, and fats) are consumed without excess, in appropriate proportions to support the energy and physiological needs, while micronutrients (vitamins and minerals) must be absorbed in relatively small quantities to ensure growth, development, metabolism, and physiological functioning [21]. These dietary goals are maintained in many diets including the Mediterranean diet, Dietary Approaches to Stop Hypertension (DASH), and Mediterranean-DASH Intervention for Neurodegenerative Delay (MIND) [22,23]. In recent years the traditional Asian diets have been added to the group of healthy diets [24]. The Mediterranean diet is based on the consumption of unrefined cereals, legumes, and the high consumption of vegetables and fruits of different colors and textures with a high content of micronutrients, fibers, and phytochemicals. Moderate consumption of animal protein (fish, white meat, and eggs) is recommended, while red meat and processed meat are rarely consumed and then in small quantities. Dairy products, recommended as a source of calcium and necessary for the health of the bones and heart, must be consumed in moderation. In the Mediterranean diet, olive oil serves as a primary source of dietary lipids. In addition, it is recommended to drink water (1.5–2 l/day) as the main source of hydration, while wine is allowed in moderation, to be consumed at meals [25]. The DASH diet is based on a model that aims to keep blood pressure, cholesterol and triglycerides low. The main features are a high consumption of fruits and vegetables, the intake of low-fat dairy, and a reduced amount of saturated and total fat and cholesterol. The DASH diet has been shown to reduce cardiovascular risk factors such as the onset of coronary artery disease, stroke, heart failure, metabolic syndrome, and diabetes [26,27].

The MIND diet can be defined as a cross between the Mediterranean and DASH diets and aims to support cognitive health during advanced age. The MIND diet is based on increased intake of fruit, fresh vegetables, beans, whole grains, fish, poultry, olive oil, and wine in moderation. In addition, foods considered unhealthy for the brain, including red meats, butter/margarine, cheese, pastries, sweets, and fried or fast food, are greatly limited. Interestingly, adherence to the MIND diet reduced the risk of developing Alzheimer’s disease by 35% [28]. Foods considered healthy or unhealthy in Mediterranean diet, NASH, and MIND are different, and we must deepen our specific knowledge of these dietary plans to fully understand their differences [29]. Among traditional Asian diets, the Korean diet is based on consumption of rice and other whole grains, fermented foods, indigenous land and sea vegetables, mainly legume and fish proteins compared to red meat, medicinal herbs (e.g., garlic, green onions, ginger), and sesame and perilla oils [30]. Unlike western diets, the Korean diet is founded on small portions, derived from seasonal food sources, and has an absence of fried foods. Epidemiological studies have shown that the relevance to this diet is related to a reduced risk of metabolic syndrome, diabetes, obesity, and hypertriglyceridemia [31]. The traditional Chinese diet mainly includes the consumption of rice or noodles, soups, vegetables, steamed bread or fruit and vegetables, soy, seafood, and meat [32]. Despite this diet being richer in carbohydrates, as it contains less fat than a western diet, the traditional Chinese diet does not seem to promote weight gain, suggesting that the restriction of carbohydrates may not be the only intervention applicable to combat obesity and cardiometabolic risk [33]. Finally, the traditional Japanese diet is characterized by small portions of several components, including rice, fish, soups, pickles, algae, fruits, vegetables, and mushrooms. Adhesion to a traditional Japanese food model has been associated with favorable effects on blood pressure and a lower prevalence of hypertension [34,35].

Beside the differences found in the diets described, the common denominator seems to be the high consumption of fruit and vegetables that is constantly present, and which guarantees a reduced onset of cancer [36,37]. To date, it is known that several natural bioactive compounds possess anticancer properties [38,39]. It is important to highlight that there are more than 100 natural plant-based compounds currently in clinical use as anticancer drugs [40].

Natural compounds with anticancer properties are capable of killing transformed or cancerous cells without being toxic to healthy cells. Most fruits and vegetables consumed with food are made up of bioactive molecules belonging to the family of polyphenols, a group of natural compounds widely distributed in the plant kingdom; this group is varied, and to date, more than 8000 phenolic structures are known [41]. Polyphenols are classified according to chemical structure, and their subdivision is represented in Figure 2.

The main natural compounds currently used against several types of cancer are shown in Table 1. In this review, among the various natural compounds with anticancer activity, four bioactive extracts (bergamot, oleuropein, curcumin, quercetin) will be discussed.

Bergamot (Citrus bergamia Risso et Poiteau) is a citrus fruit growing almost exclusively in the south of Italy in a restricted area of the Calabrian coast, thanks to the particular environmental conditions that are favorable for its cultivation. This citrus fruit is defined as a hybrid between a sour orange (*C. aurantium* L.) and lemon (*C. limon* L.) or between a sour orange and lime. Bergamot possesses a unique profile of flavonoids and flavonoid glycosides [91]. To date, several important properties are recognized in bergamot, including antioxidant, anti-inflammatory [92,93,94,95,96,97], neuroprotective, hypoglycemic, and hypolipemic properties against many metabolic diseases [98,99,100]. Bergamot fruit is mostly used for its essential oil (BEO), obtained by rasping the fruit peel, its polyphenolic fraction (BPF), and its juice (BJ), which is obtained by squeezing the fruits and was initially considered as a secondary product.

Olive oil is one of the main constituents of the Mediterranean diet and is extracted from olive drupes that contain known biophenol secoiridoids. Among these, one of the best known is oleuropein, which was proven to possess numerous beneficial properties including antioxidant, anti-inflammatory, anticancer, antiviral, hypoglycemic, neuroprotective, and antiaging effects [101,102,103]. For this reason, this natural compound is considered to be a “super functional food”.

Turmeric (Curcuma Longa) is a plant belonging to the ginger family (Zingiberaceae), native to India but present also in China, Southeast Asia, and Latin America. Turmeric is a common spice, but in recent decades has aroused scientific interest for its therapeutic potential with anticancer, anti-inflammatory, antidiabetic, antiaging, and neuroregenerative effects [104,105,106]. Curcumin, a yellow pigment from Curcuma longa, is the major component of turmeric, and chemically it is a poliphenol.

Quercetin is a flavonoid ubiquitously present in fruits and vegetables, and for this reason its intake is very common in the Mediterranean diet. It has also been recognized that quercetin performs numerous beneficial functions by acting as an antioxidant, anticarcinogenic, anti-inflammatory, antidiabetic, and antimicrobial [107,108]. Quercetin demonstrates dose-dependent effects: at low concentrations, it acts as an antioxidant, while at high concentrations, it is a pro-oxidant compound [109].

Currently, knowledge and experience regarding the anticancer activity of natural compounds is increasing.

Since the treatment of cancer with a known chemotherapy induces the onset of systemic side effects, such as cardiotoxicity or nephrotoxicity, which frequently require the early withdrawal or replacement of treatment, the use of natural compounds as adjuvants of chemotherapy could reduce the necessary doses and treatment times. In this way, the desired anti-proliferative effect could be achieved by reducing the possibility of developing systemic side effects. The use of natural compounds has increased exponentially in the last decades as it is known that plant extracts do not involve side effects at the systemic level. Nevertheless, it is key to note that it is important that studies in the scientific literature have been conducted in accordance with best practices of pharmacological research so as not to disclose information that is only partially correct [110,111,112].

### 2.1. Natural Compounds and Cancer: Cellular Viability

The use of natural compounds with anticancer effects has increased thanks to their low toxicity and lower side effects, which allow their use in the treatment or adjuvant therapy of cancer. Apoptosis is programmed cellular death, finely regulated at the gene level, resulting in efficient removal of damaged cells. Induction of apoptosis is crucial in precancerous lesions since harmful cells are eliminated by preventing uncontrolled cell proliferation and cancer progression. Deregulation of apoptosis is considered one of the characteristics of cancer progression, and transformed cells are able to circumvent this process, although the mechanisms involved are not sufficiently known. For this reason, therapeutic strategies aimed at restoring the sensitivity of cancer cells to apoptosis are increasingly tested [113,114,115]. Citrus fruits represent major sources of flavonoids. Several experimental studies have strongly indicated that bergamot and its extracts can exert antitumor effects thanks to the ability of flavonoids to interfere with the main stages of carcinogenicity: the onset, promotion, and progression of cancer [116]. The anticancerous action of BEO has been adequately highlighted in several in vitro works. In particular, a reduction in cell proliferation was triggered by the shutdown of the cell cycle in phase G0–G1. In addition, intense pro-oxidant activity and cellular DNA damage have been appreciated [80,116,117]. A very comprehensive work [118] conducted in vitro on human cancerous cells of the nervous system (SH-SY5Y, PC12), prostate (PC3), and breast (MDA-MB-231) showed that treatment with BJ at different concentrations (1–5%) arrested cancer progression. In addition, BJ demonstrated its ability to reduce the growth rate of various cancer cell lines with mechanisms dependent on the type of cancer [119,120]. Finally, it has been shown in human colon cancer cells that low concentrations of BJ can induce inhibition of the mitogen-activated protein kinase (MAPK)-dependent pathways, and cause cell cycle arrest and alteration of apoptosis, while high concentrations produce oxidative stress, causing DNA damage [121]. BPF has attracted scientific attention for its peculiar composition and high content of flavonoids, such as naringin, hesperidin, and neoeriocitrin [79]. Although few studies on the correlation between BPF and cancer are available, multiple papers indirectly involving BPF are known. In fact, cholesterol-lowering drugs are able to reduce cancer incidence and cancer-related mortality [122]. To date, it is known that BPF possesses several hypolipidizing properties against many metabolic dysfunctions [123,124,125,126].

Oleuropein is the polyphenol most present in olives and olive leaves, and its anticancer properties are well-known for several types of cancer, including breast, lung, liver, prostate, and colon [127,128,129,130]. Belonging to the secoiridoids, a group of compounds found exclusively in all 500 species of Oleaceae plants, oleuropein results in a reduction of cell proliferation mainly by using two mechanisms: on the one hand, it acts by stopping the cell cycle through the upregulation of cyclin-dependent kinase (CDK) inhibitors, and on the other, by modulating the genic expression responsible for the induction of intrinsic and extrinsic pathways of apoptosis through the upregulation of p53 and p21. In addition, oleuropein can alter the function of key molecules involved in the onset and development of cancer such as MAPKs, c-Met proto-oncogene, and the fatty acid synthase (FASN) enzyme [131]. A recent study [132] conducted in vitro on two genetically different triple-negative breast cancer (TNBC) cell lines (MDA-MB-231 and MDA-MB-468) demonstrated the ability of oleuropein to suppress cell proliferation, stimulate apoptosis through S-phase cell cycle arrest, and express the initiating caspases of the apoptotic process (Caspase1, 4, and 14). A similar result was also obtained for differentiated thyroid cancer unresponsive to the current radioiodine-based treatment [133]. In this work, a reduction in the proliferation of the TPC-1 and BCPAP lines of thyroid cancer was demonstrated, while only mild effects were detected in the non-tumor thyroid TAD-2 cell line. Once again, the mechanism involved in the reduced proliferation was a dose-dependent S-phase cycle arrest. In this paper, the effect of peracetylated oleuropein, obtained by peracetylation reactions that improved the stability of oleuropein and its ability to permeate within the cells [134,135], was also evaluated and compared. The peracetylated oleuropein, as expected, responded better than oleuropein, demonstrating stronger stoppage in the S-phase of the cell cycle. There are several studies that describe the role of olouropein in reducing the expression of histone deacetylase II (HDAC2), HDAC3, and HDAC4, thus inducing apoptosis but also delaying cell migration and invasion [136]. Due to the low concentration of phenolic secoiridoids in the main foods of our diet, their reduced absorption, and rapid metabolic transformation, it is difficult to obtain any therapeutic potential from their consumption in food alone. In addition, difficulties in many human models are also associated with the enormous variability of nutraceuticals in chemical terms, composition, and preparation, as well as in the quantification of the dosage and in the choice of appropriate formulations to be administered. For this reason, in vivo biological effects and human trials still require further investigation. To date, it is possible to use oleuropein as an adjunct to conventional cancer therapies: recent studies have shown in animal models that the addition of oleuropein to cisplatin protected against the toxic effects generated by the drug [137,138].

Quercetin is a flavonoid compound normally present in nature in a variety of plants, fruits, and vegetables that are consumed daily in the diet. The numerous beneficial anticancer, antioxidant, and anti-inflammatory effects of quercetin have already been amply demonstrated. The antitumor effects of quercetin, observed both in vitro and in vivo, are related to its ability to alter cell cycle progression, promote apoptosis, inhibit cell proliferation, inhibit the progression of metastases, and angiogenesis [139]. Several in vitro studies have highlighted the antitumor role of quercetin: for example, in ovarian carcinoma (SKOV3 cell line), quercetin induced a decrease in cyclin D1, with consequent arrest in the S and G2/M phases of the cellular cycle. In human leukemia (U937 cell line), quercetin has been shown to induce cell cycle arrest at G2/M following the decrease in cyclins D, E, and E2F, and in osteosarcoma cells (HOS), quercetin was able to induce changes in the G0/G1 phase [140,141,142]. In addition, quercetin modulates the regulation of p53-related pathways, inhibiting the activity of CDK2 and cyclins A and B. Direct involvement of p53 was also demonstrated in breast cancer, where the MDA-MB-453 cell line increased the expression of this protein [143,144]. It was also demonstrated that quercetin induced apoptotic death of tumor cells (A375SM melanoma cell, HL-60 acute myeloid leukemia cell, and A2780S ovarian cancer cell), increasing the expression of pro-apoptotic proteins and decreasing the level of antiapoptotic proteins [145]. More specifically, quercetin was able to increase the release of cytochrome c from the mitochondria, activate the expression of caspase-3, -8, -9, Bax, and Bad, and downregulate the antiapoptotic proteins, including Bcl-XL, Bcl-2, and Mcl-1 [146,147]. Models of various types of cancer in vivo have also been studied, and quercetin was shown to inhibit their growth, increase the survival rate of the animals, and significantly reduce the volume of the tumor [148]. Quercetin was also proven capable of promoting apoptosis and inhibiting proliferation, angiogenesis, and metastasis. These effects were found in models of breast, pancreatic, prostate, and lung cancer; the dosage of quercetin was 50 mg/kg [149,150,151,152,153]. In the last decade, it has been shown that quercetin is able to increase its antitumor effect when the treatment is associated with other compounds. For example, the liposomal co-encapsulation of vincristine and quercetin was shown to be an improved therapy [154,155].

Curcumin is the most representative polyphenol extracted from the rhizomes of Curcuma longa, with a typical yellow color. Curcumin is notoriously used as a component in cosmetics, and as a flavoring for foods, beverages, and dietary supplements. To date, curcumin has shown numerous therapeutic benefits against inflammation, oxidative damage, obesity, metabolic syndrome, neurodegenerative diseases, and several cancers. Furthermore, all these beneficial properties are justified by the chemical structure of curcumin [156]. Curcumin was reported to prevent the growth of many tumors, inhibiting cell growth, blocking the cell cycle, and stimulating apoptotic death; for example, in the human colon cancer cell line HCT-116, it inhibited cell proliferation by cell cycle arrest at the G2/M phase and/or in a small quantity in the G1 phase [157]. In other studies, curcumin downregulated the genes for p21 and p27 (SMMC-7721 hepatoma cells) [158] or upregulated the gene for p53 (HCT116 colon, MCF-7 breast, and CNE2, 5-8F nasopharyngeal cancer cells) [159]. In addition, curcumin triggered caspase 8, 3, and 9, inevitably reaching the activation of apoptotic death [160]. The proinflammatory transcription factor NF-κB regulates more than 500 different genes expressing for proteins involved in cellular signaling pathways, so all compounds that interact with NF-κB, inhibiting it, may be used in cancer therapy. Curcumin was able to downregulate NF-κB in breast cancer cells [161] and played an important role in hematologic tumors: in leukemia, curcumin stopped nuclear translocation of NF-κB and the degradation of human myeloid ML-1a cells [162]; moreover, curcumin triggered apoptotic death in B-cell chronic lymphocytic leukemia (CLL-B) by downregulation of the STAT3, AKT, and NF-κB proteins [163]. The treatment of gastric cancer indicated the pharmacological efficiency of curcumin, which inhibited antiapoptotic proteins of the Bcl-2 family and increased the expression of caspases 3, 8, 9, p53, and Bax [164]. The scientific literature offers various examples in other types of cancer (lung, colorectal, liver, and pancreas), but the mechanisms are always related to proliferative reduction, the onset of apoptosis, and cell cycle blockage [165]. Curcumin has been tested on animal models, and the optimal dose to reduce cancer was found to be 300 mg/kg [166]. In humans, early preclinical studies were conducted to evaluate the tolerated dose, but further studies should nevertheless be conducted [167]. The data reported in this section show that the four polyphenolic compounds considered (Bergamot, Oleuropein, Quercetin, and Curcumin) have the ability to reduce cell viability in cancer. To achieve this, natural compounds are able to stop the continuation of the cell cycle, induce apoptotic death, and increase the expression of the tumor transcription factor p53. These mechanisms are displayed in Figure 3.

### 2.2. Antioxidant Effect of Natural Compounds on Cancer

The cell metabolism also includes oxidative reactions useful for ensuring survival; in fact, reactive oxygenated/nitrogenated species are produced and involved in several regulatory processes, including gene expression, cell proliferation, and apoptosis. When reactive species are generated in excess of the cellular antioxidant capacity, the main biological molecules—such as DNA, proteins, lipids, carbohydrates, and enzymes—may be oxidized, losing and/or altering their functions [168]. For example, reactive oxygen species (ROS) interact extensively with nuclear DNA, generating mutations and genomic instability; with proteins, generating protein adducts; with the lipids of cell membranes, altering their functions. This damage induced by oxidative stress has been found in pathological conditions and cancer cells [169]. To counteract reactive species, endogenous antioxidants are physiologically present in the body. However, if the endogenous compounds cannot provide complete protection, there are also exogenous antioxidants provided by food, food supplements, and pharmaceuticals. In particular, the capability of natural products to reduce cellular oxidative stress has been investigated in recent years [170]. Mitochondria are important organelles of the eukaryotic cells, playing an essential role in energy metabolism. Mitochondrial dysfunction is known to be associated with cancer. Free radicals produced by mitochondria as products of their normal metabolism can include the hydroxyl radical (OH•), superoxide anion (O_2_•−), hydrogen peroxide (H_2_O_2_), hydroxyl ion (OH−), and nitric oxide (NO•). Most natural compounds have antioxidant activity owing to the presence of one or more catechol groups in their structure, which are responsible for eliminating reactive oxygen species, thus inhibiting the formation of free radicals and lipid peroxidation [171,172,173]. The catechol also known as 1,2 dihydroxybenzene is characterized by a brute formula equal to C_6_H_6_O_2_, and two hydroxyl groups are placed in the ortho position on the benzenic ring [174].

Scientific literature has amply demonstrated that bergamot fruit has a robust antioxidant property, and for this reason, its consumption is encouraged as health-promoting. For example, BJ has been shown in vitro to possess a significant antiradical property against superoxide and nitric oxide, O_2_• scavenging activity, and lipid peroxidation inhibition. Parallel studies conducted in vivo on subjects fed hearts of mice with BJ or vehicle for three months showed statistically significant antioxidant responses [76]. Naringenin, a polyphenol belonging to the class of flavanones and widely distributed in citrus fruits, is one of the major components of BPF [175]. In fact, this compound has been shown to induce cytotoxic and apoptotic effects and prevent cell proliferation in different types of cancer cells [176,177,178]. Unfortunately, its practical use in vivo is reduced owing to its hydrophobic nature, short half-life, and poor absorption. For this reason, the use of nanomaterials was suggested to improve its bioavailability [179].

Numerous scientific studies have highlighted the antioxidant properties of oleuropein and its ability to promote the activity of ROS-detoxifying enzymes, including superoxide dismutase (SOD), catalase (CAT), glutathione S-reductase (GSR), and glutathione S-transferase (GST). In addition, this compound inhibits lipid peroxidation, and the antioxidant role of oleuropein is beneficial in different types of cancerous processes [180]. There are several studies describing how oleuropein, in addition to reducing ROS, decreased the expression of histone deacetylase, induced apoptotic death, and delayed the migration and invasion of cells in a dose-dependent manner. In addition, oleuropein induced the downregulation of metalloproteinase genes, which may be involved in the prevention of breast cancer metastases [113,116,181]. An antioxidant and growth inhibitory effect was found for differentiated thyroid cancer [115].

Curcumin has also been shown to possess, both in vitro and in vivo, a strong antioxidant activity, and through this peculiarity, is able to reduce some stages of cancer progression. In addition to increasing the activity of many antioxidant enzymes such as SOD, CAT, GST, and GSR, curcumin inhibits the direct formation of reactive species including superoxide radicals, hydrogen peroxide, and nitric oxide radical [182]. It has been shown that curcumin is also able to increase the activity of detoxifying enzymes in the liver and kidneys, reducing xenobiotics and protecting against carcinogenic processes [183]. Curcumin also played a radioprotective role and modulated the malondialdehyde levels in a lung carcinogenesis model induced by benzo(a)pyrene, a major carcinogenic pollutant, in mice [184]. Finally, it is important to note that curcumin also prevents brain injury, thanks to the suppression of oxidative stress via the AKT/nuclear factor-E2-related factor 2 pathway, then acting as a neuroprotector [185]. Quercetin acts as an antioxidant by reducing high-valent iron and thereby inhibiting lipid oxidation and the production of iron-catalyzed ROS; in addition, it regulates signal transduction pathways such as NRkB, MAPK, and AMPK [186]. Due to its poor toxicity, quercetin was shown to possess various inhibitory effects on many steps of carcinogenicity [187]. The antioxidant effect has been carried out, by polyphenols of interest, thanks to the presence of the catechol group and the increased expression of antioxidant enzymes. These mechanisms are displayed in Figure 4.

### 2.3. Anti-Inflammatory Effect of Natural Compounds on Cancer

Inflammation was associated with the development and progression of cancer by the end of the 19th century, owing to the discovery of leukocytes in neoplastic tissues. Yet the clear evidence that inflammation plays a critical role in tumorigenesis is relatively recent, and over the past 10 years this correlation has begun to have implications for cancer prevention and treatment [188]. Currently, the correlation between inflammation and cancer is explained by two pathways that can occur: the intrinsic pathway, in which genetic events determine the formation of neoplasia and the subsequent and consequent construction of an inflammatory microenvironment; and the extrinsic pathway, which starts with an inflammatory process that, after becoming chronic, facilitates the development of cancer [189]. Chronic inflammation is characterized by prolonged tissue damage, in which cell proliferation is induced for the purpose of repairing damaged tissues. This phenomenon, known as “metaplasia”, is normally a reversible process that lasts only for the time necessary to physiologically reconstitute the damaged segment. In some circumstances, metaplasia can also turn into “dysplasia”, a phenomenon that involves a disorder of cell proliferation and leads to the production of atypical cells; frequently, dysplasia is the event preceding cancer formation [190]. The chronic inflammatory microenvironment is characterized by a cellular component (macrophages, leukocytes, and dendritic cells) and a molecular component (proinflammatory cytokines, chemokines, adhesion molecules, and inflammatory enzymes). The combination of both components generates the binomial cancer inflammation [191]. The anti-inflammatory activity of bergamot derivatives has been demonstrated in both in vitro and in vivo studies: for example, BEO reduced carrageenan-induced inflammation in rats, an effect that was measured as a reduction of paw volume [192]. In addition, BEO reduced the levels of the mRNA of IL-8 in cells treated with TNF-α. Graziano et al. highlighted a significant decrease in skin inflammation with a reduction of intercellular adhesion molecule 1 (ICAM-1), with inducible nitric oxide synthase (iNOS), nitric oxide (NO), and ROS after consuming BJ [193]. In addition, BJ was shown to reduce certain inflammatory cytokines (IL-1β, IL-6, TNF-α, NF-κB) in activated monocytes [194]. Impellizzeri et al. determined that BJ reduced the levels of IL-1β, TNF-α, nitrotyrosine, p-JNK, ICAM-1, P-selectin, and NF-κB in an inflammatory model of colitis [195]. It is also important to mention the scientific work of Currò et al., which highlighted an anti-inflammatory effect of BJ and a reduction in IL-1β, IL-6, and p-JNK in a model of neuroinflammation [196]. Another study by Nisticò and collaborators highlighted the ability of BPF to reduce UVB-induced photoaging in immortalized human keratinocytes. In particular, the expression of inflammatory cytokines, changes in telomere length, and cell viability were examined. The results showed that BPF modulates the transduction pathways of the basic cellular signal, leading to antiproliferative, antiaging, and immunomodulating responses [82]. Navarra et al. showed in vivo that BJ generated a significant dose-dependent reduction in preneoplastic lesions of the colon. Additionally, a downregulation of inflammation-related genes (COX-2, iNOS, IL-1β, IL-6, and IL-10) was shown in rats taking BJ [197].

The protective effects of oleuropein against inflammation are multiple: in vivo preliminary studies demonstrated a significant anti-inflammatory effect generated by oleuropein during lipopolysaccharide-induced sepsis (LPS) in mice. To study an induced inflammatory effect, LPS has been widely used in both in vitro and in vivo scientific work [198,199]. In fact, pretreatment with oleuropein ameliorated LPS-induced liver and kidney histological changes, mitigated the increased levels of malondialdehyde, and reduced the levels of reduced glutathione and the number of inflammatory biomarkers (TNF-α, IL-1β, and IL-6) [200]. Scientific works already published have highlighted the protective role of oleuropein in several cancer cell lines, including leukemia, breast, pancreatic, prostate, and colorectal [201,202,203]. It is important to point out that oleuropein proved capable of discriminating between cancer and normal cells, inhibiting proliferation and inducing apoptosis only in cancer cells [204,205,206]. Oleuropein’s mechanism is downregulation of proinflammatory enzymes IL-6 and interleukin 1β [207,208].

The anti-inflammatory effect of curcumin is mainly related to its ability to inhibit the activity of certain enzymes directly involved in inflammatory disorders and cancer, such as cyclooxygenase-2 (COX-2), lipoxygenase (LOX), and inducible nitric oxide synthase (iNOS). In fact, improper regulation of these enzymes has been associated with the onset of pathophysiological disorders [209]. In addition, curcumin can suppress proinflammatory pathways, blocking both tumor necrosis factor alpha (TNF-α) production and cell-mediated signaling from TNF-α in various cell types. Both in vitro and in vivo studies have shown that curcumin can direct block TNF-α, binding to this molecule and deactivating it [210]. Growing evidence has shown that curcumin exerts an interesting anticancer property: for example, several studies demonstrated that curcumin (12 g/day for three months) induces antiproliferation and apoptosis in several cancer cell lines such as breast, pancreatic, prostate, kidney, and colorectal [211]. Curcumin also acts in the regulation of transcription factor NF-κB, the expression of which is associated with the progression of several types of cancer. In fact, NF-κB can be induced by carcinogens, free radicals, endotoxins, cytokines, and ionizing radiation. Specifically, curcumin acts as an NF-κB regulator, suppressing the activation of IκB kinase (IKK), which is responsible for the nuclear translocation and activation of NF-κB [212]. Due to its anti-inflammatory action, curcumin is expected to exert chemopreventive effects on carcinogenesis. Emerging preclinical evidence has pointed out that to reduce the side effects of prolonged treatment with chemotherapy, it is advisable to use combined therapies that promote anticancer efficacy without increasing the toxicity [213]. Docetaxel, a chemotherapeutic agent belonging to the class of taxan drugs, is used for the treatment of several neoplasms, in particular, for breast cancer, lung cancer, prostatic carcinoma, and gastric adenocarcinoma. Banerjee et al. highlighted that combined treatment with docetaxel (10 nM) and curcumin (20 µM) for 48 h significantly inhibited cellular proliferation and induced apoptosis in prostate cancer, compared to curcumin and docetaxel alone [214]. 5-Fluorouracil (5-FU) is considered a highly important chemotherapeutic drug and has been widely used in the treatment of colorectal cancer. Unfortunately, patients treated with this drug often develop a high resistance to it. The combination of 5-FU and curcumin could overcome these difficulties, however, and pretreatment with curcumin (5 µM)-enhanced 5-FU (0.1 µM) chemosensitization reversed the resistance [215]. Cisplatin, an inorganic platinum agent that can induce DNA–protein crosslinks, is widely used as a standard therapy for metastases and advanced bladder cancer. However, almost 30% of patients do not respond to initial chemotherapy. Co-treatment with curcumin (10 µM) and cisplatin (10 µM) displayed a powerful synergistic effect, causing the activation of caspase-3 and overregulating phospho-extracellular signaling of 1/2 Kinase (p-ERK1/2) compared to curcumin or cisplatin alone [216]. In addition to these described effects, the implication of curcumin in combination chemotherapy has been tested in several clinical trials.

Quercetin has a strong and long-lasting anti-inflammatory capacity; several in vitro studies using different cell lines have shown that quercetin inhibits LPS-induced TNF-α accumulation in macrophages and the production of LPS-induced IL-8 in A549 lung cells. Quercetin inhibits the production and activity of enzymes that produce inflammation COX and LOX [217], limits inflammation induced by LPS by inhibiting phosphatidyliinositol-3-kinase (PI3K), and inhibits the release of proinflammatory cytokines. A study carried out on human umbilical vein cells (HUVEC) showed a protective effect of quercetin against inflammation induced by H2O2 and indicated that this effect was mediated by the sub-regulation of adhesion molecule 1 (VCAM-1) in vascular cells [218]. Quercetin also affected immunity and inflammation in vitro by acting directly on leukocytes and modulating many intracellular signaling kinases [219]. Several studies have shown that quercetin decreased the histological signs of acute inflammation by suppressing leucocyte recruitment, decreasing chemokine levels, and stopping lipid peroxidation in an experimental rat model [220]. There are several studies in humans that have supported the antipathogenic capacities of quercetin. The co-ingestion of two or more flavonoids increases their bioavailability, which affects immunity and inflammation. In particular, when taken together, quercetin showed a successful reduction in illness rates [221]. In addition to the anticancer activity of quercetin as demonstrated by the induction of apoptotic death and the arrest of the cell cycle, this natural compound also acts on the process of angiogenesis and formation of metastases in cancer cells. It was shown in breast and prostate cancer that quercetin exerts an anticancer action by inhibiting the growth of blood vessels by suppression of the vascular endothelial growth factor-2 (VEGFR-2), an important signaling protein involved in angiogenesis [222]. In addition, quercetin can also inhibit the onset of metastases by modulating the expression of caderins, the molecules that mediate cellular adhesion under conditions where the inflammatory process is switched off [223]. The anti-inflammatory effects of Bergamot, Oleuropein, Quercetin and Curcumin work by inhibiting the cytokines and cells involved in the inflammatory process. These mechanisms are displayed in Figure 5.

### 2.4. Bioavailability of Polyphenols

The bioavailability of any compound taken with food varies in relation to its digestion, absorption, and metabolism. Precisely for this reason, there is no correlation between the amounts of compounds consumed in food and their bioavailability in the human body.

Polyphenols, once ingested, must also be absorbed and transformed into bioactive compounds. In general, after ingestion, polyphenols meet an enzymatic cleavage of the carbohydrate portion (when present) and its aglycones enter the epithelial cells of the small intestine through passive diffusion [224].

If polyphenolic compounds cannot be absorbed in this district, they reach the colon where they are metabolized by the microbiota. It is, therefore, likely that an alteration of the gut microbiota will contribute to a reduction of polyphenol absorption and worsening of human health [225,226]. However, it should also be noted that polyphenols could have beneficial effects on the composition of the gut microbiota, acting as prebiotics [227]. Although the molecular mechanisms by which polyphenols can behave as prebiotics have not been fully clarified, it is assumed that they perform selective antimicrobial activity against pathogenic bacteria [228].

Subsequently, the final derivatives absorbed are conjugated by methylation, sulfation, and glucuronidation reactions, and reach the liver through the blood circulation, where may be subjected to phase II metabolism, transported into the appropriate tissues, and finally, excreted by urine or feces [229]. With this information, we can conclude that the beneficial effects of polyphenols for health depend on both the amount taken and the bioavailability [230]. In fact, a certain quantity of polyphenols should be consumed through the diet so that the concentrations of their metabolites present in the blood are not too low, to ensure their beneficial effects. The bioavailability of polyphenols varies between the different classes and depends on the chemical structure. The studies conducted have allowed us to build a scale of bioavailability for the different polyphenols that, in order of size from the largest to the smallest, can be reported as: phenolic acids > isoflavones > flavonols > catechins > flavanones, proanthocyanidins > anthocyanins [231,232,233]. Because of this characteristic of polyphenols, it would be interesting to standardize their extraction, so as to know the amount of the initial intake, and to quantify the main metabolites present before their elimination. This way, we would have better knowledge of the quantities of polyphenols needed to ensure the performance of their activities.

### 2.5. Epigenetics, Cancer, and Involvement of Polyphenols

Epigenetics deepen the changes that can occur in DNA, affecting gene expression. These variations are heritable from cell to cell, and once established, are relatively stable [234]. Although these changes occur during the early development of embryonic and primordial cells, it is now known that they can also occur over the course of life; the main causes may be drug use, incorrect diet, or exposure to an unfavorable environment [235]. The various epigenetic mechanisms work in a concerted and interdependent manner to regulate gene expression.

Epigenetic alterations often occur in the early stages of cancer development and cancer cells; therefore, may inadequately activate oncogenes or inactivate tumor suppressors. For this reason, being able to prevent epigenetic alterations can reduce the proliferation of cancer cells, the severity of cancer growth, and metastases [236,237].

The major epigenetic modifications studied include DNA methylation, covalent histone modification, and non-coding RNA modification. DNA methylation is a simple addition of a methyl group (CH_3_) in position five on the pyrimidine ring of the cytosine residue in a cytosine-guanine pair (CG) and is essential for life. Nevertheless, this reaction on CpG dinucleotides is involved in many point mutations that lead to genetic diseases in humans and that increase the mutagenic risk [238]. In cancer, DNA hypomethylation can lead to genome instability, DNA rupture, and the promotion of uncontrolled growth. On the contrary, DNA hypermethylation usually leads to gene silencing, which is one of the most common somatic aberrations in cancer [239].

Possible histone modifications include acetylation, citrullination, ubiquitylation, deamination, mono-, di-, and tri- methylation, phosphorylation, and ribosylation. These changes can affect chromatin density and result in a change in DNA accessibility, altering gene regulation. It has been shown that most of these changes are found in many tumors [240]. Small non-coding RNAs include microRNA (miRNA), small-interfering RNA, and small nucleolar RNA, which intervene in various biological processes such as development, metabolism, and maintenance of homeostasis. The main role of miRNA is to participate in the regulation of gene expression through translation inhibition and heterochromatin formation, while their regulation is controlled by epigenetic effects. miRNA genes function as a genome surveillance mechanism; they can be epigenetically regulated by DNA methylation or specific histone modifications. Dysregulation of miRNA is associated with the development of numerous metastatic tumors [241]. It is widely accepted that many cancers could be avoided by changing the food model and lifestyle. In addition, it has been demonstrated that food may have epigenetic effects, and polyphenols contained in fruit and vegetables are included as part of that [242]. Therefore, their consumption is also highly recommended to understand the implications of diet on epigenetic modifications and cancer development. Examples are given below: Resveratrol is a natural polyphenol that is found in peanuts and in the peel of grapes and berries. The epigenetic effects of resveratrol have been evaluated in numerous cancer models. Resveratrol was able to induce dysregulation of numerous miRNA in cancer cells, but not in healthy ones [243]. In addition, this natural compound can protect against cancer by regulating transcriptional suppression of a number of genes, including p53 [244]. Green tea polyphenols have been shown to inhibit the tumor, its invasion, and angiogenesis in a mouse model of skin cancer [245]. In particular, Epigallocatechin gallate (EGCG), a bioactive polyphenol in green tea, has demonstrated epigenetic effects in humans through the demethylation of the promoters of some tumor suppressor genes [246].

Soybean isoflavones can influence the onset of cancer, and it has been demonstrated that soybean intake during childhood and adolescence has reduced the risk of breast cancer [247]. The methylation of five cancer-related genes was evaluated in menopausal women, and was evidenced that consuming a mix of isoflavones induced increased methylation in some genes associated with breast cancer development, and reduced the risk of developing the disease [248].

In addition to the examples given, numerous other compounds belonging to the group of polyphenols perform epigenetic functions: it is important to mention lycopene, curcumin, quercetin, isothiocyanates, genistein, and caffeic acid, among others [249,250,251,252].

## 3. Discussion

Although cancer is one of the world’s leading causes of death, a more optimistic view for the future stems from the awareness that there have been many improvements in diagnosis and treatment approaches. In particular, early detection can address the disease with more satisfactory results, and less invasive treatments aim to increase tolerability in patients [253]. The ultimate goal is to reduce the mortality rate of cancer patients by increasing the expectation of quality of life. A total of 90% of cancers are attributable to modifiable risk factors, including a non-optimal diet, environmental pollution, excessive body weight, consumption of alcohol or tobacco smoke, physical inactivity, and infectious agents [254]. The diet is an essential element for maintaining health, and it has been estimated that bad eating habits are responsible for 5–10% of total cancer cases [255]. Several studies in the scientific literature reported that the healthiest diet is the Mediterranean diet, which is based on the consumption of high quantities of fruit, vegetables, dried fruits, legumes, cereals, fish, and extra virgin olive oil, a moderate amount of wine and small amounts of red meat, eggs, and dairy products [256]. Numerous clinical and epidemiological studies have shown that the Mediterranean diet is protective against the onset of many diseases such as diabetes, obesity, cardiovascular diseases, and cancer [257]. Most foods of plant origin belong to the class of polyphenols, the largest group of phytochemicals, that are proven to play an important role in the prevention of various diseases, including cancer, cardiovascular diseases, diabetes, and degenerative neurodegenerative diseases [258].

In this review, the anticancer properties of four of these compounds were explored: bergamot, oleuropein, curcumin, and quercetin. As reported in the literature, the natural compounds considered have numerous protective effects and tend to reduce altered physiological conditions, respecting cellular homeostasis. Conversely, in the case of cancer, the main purpose is to use substances that may be harmful to cancer cells by exerting an antiproliferative and pro-apoptotic effect to block their growth [259,260]. A suitable anticancer drug is a molecule able to distinguish specifically between healthy and transformed cells, so as to be harmless to the first and harmful to the second. Since a molecule with these characteristics is not yet available, the scientific community is investigating the action of natural compounds, which generally produce fewer side effects than conventional drugs. Due to concomitant events possibly amplifying tumor transformation and growth, such as the inflammatory process and oxidant activity, it is necessary to find a molecule with antiproliferative action and at the same time anti-inflammatory and antioxidant properties [261,262]. The compounds considered and explored in this review (bergamot, oleuropein, curcumin and quercetin) have demonstrated exactly this behavior in vitro and in vivo. In particular, cell growth was reduced, triggering antiproliferative and/or death pathways. At the same time, antioxidant and anti-inflammatory effects were noted, reflecting the chemical structure of these natural compounds and preventing the addition of characteristics to the tumor that would exacerbate it. It was also interesting to note that some mechanisms of these natural compounds acted selectively on cancer cells but not on their healthy counterparts.

## 4. Conclusions

In light of the results reported in this review, it will be interesting to increase the pre-clinical data on the use of these substances in anticancer therapy. Although there is little information about the bioavailability of natural compounds in vivo, the absence of systemic side effects developed by polyphenols, and their epigenetic involvement in cancer biology, make them particularly interesting and encourage scientific research to deepen the information available, so that they can be considered a valid support in anticancer therapy.

It is important to emphasize, however, that further high-quality studies are needed to clearly demonstrate the clinical efficacy of plant extracts.

## Figures and Tables

**Figure 1 nutrients-13-03834-f001:**
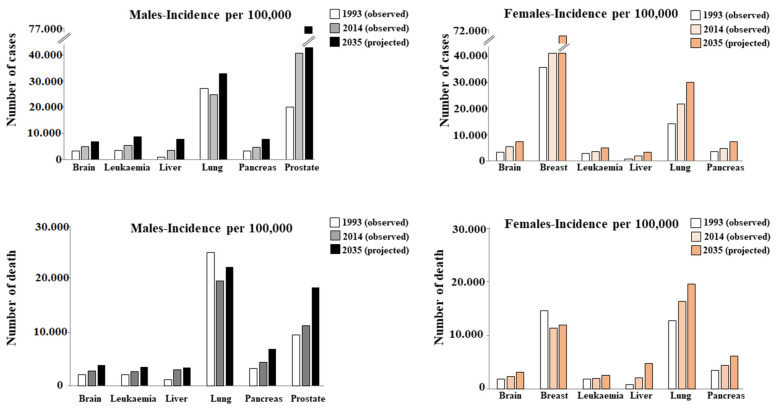
Incidence and mortality of six types of cancer. Taken and modified from [3].

**Figure 2 nutrients-13-03834-f002:**
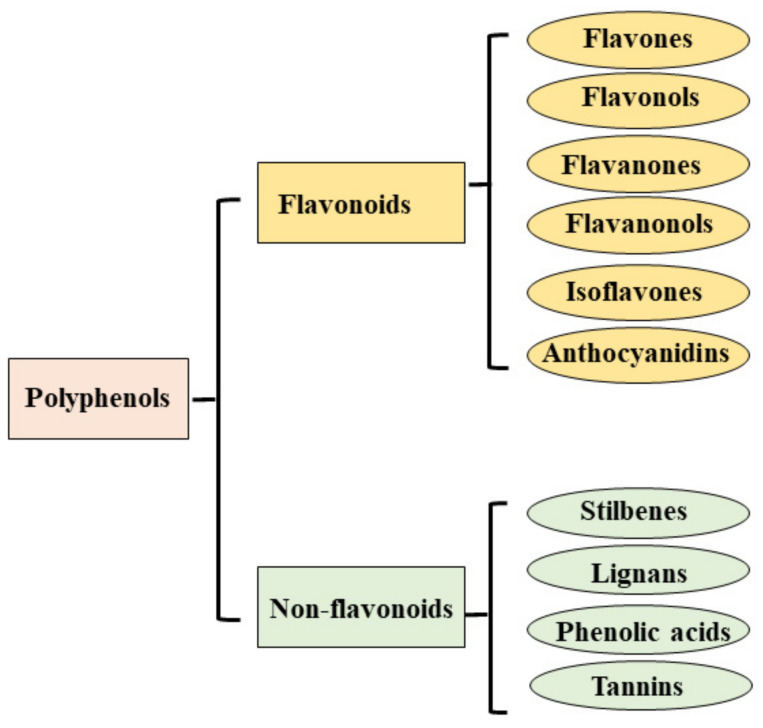
Classification of polyphenols.

**Figure 3 nutrients-13-03834-f003:**
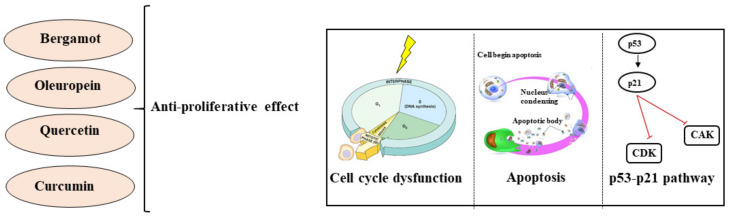
Anti-viability effects exerted by bergamot, oleuropein, quercetin, and curcumin.

**Figure 4 nutrients-13-03834-f004:**
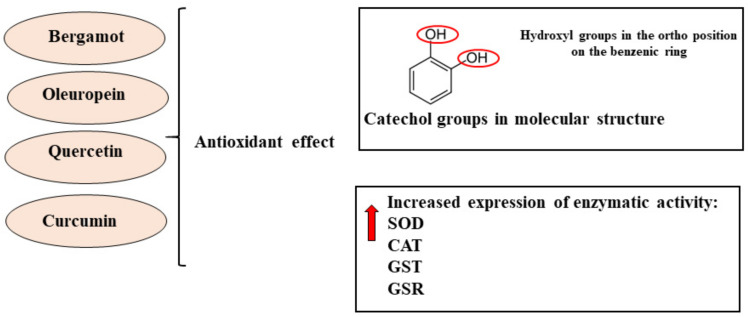
Antioxidant effects exerted by bergamot, oleuropein, quercetin and curcumin.

**Figure 5 nutrients-13-03834-f005:**
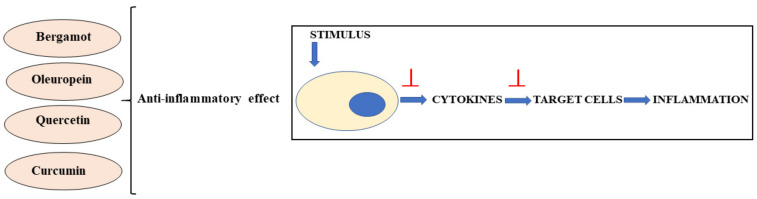
Anti-inflammatory effects exerted by bergamot, oleuropein, quercetin and curcumin.

**Table 1 nutrients-13-03834-t001:** Natural compounds with anticancer activity.

Target Cancer	Compounds	Source	Biological Activity	Ref.
**Breast**	Fucoxanthin	*Marine carotenoid*	Anti-proliferative	[42]
	Punicalagin	Pomegranate juice	Apoptosis	[43]
	Resveratrol	Grape skin and seeds	Apoptosis	[44]
	Epigallocatechin-3-gallate	Green tea polyphenols	Antiangiogenic	[45]
	Sulforaphane	Cruciferous vegetables	Apoptosis	[46]
	Genistein	Soy	Phytoestrogen	[47]
	All-trans-retinoic-acid	Vitamin A	Apoptosis	[48]
	Parthenolide	*Tanacetum parthenium*	Apoptosis	[49]
	Soy	Vegetarian food	Antiangiogenic	[50]
	Garlic	*Allium sativum*	Apoptosis	[51]
**Lung**	Apigenin	Flavonoids	Anti-proliferative	[52]
	Lupeol	Guttiferae	Anti-proliferative	[53]
	Saponin	Soapwort plant	Apoptosis	[54]
	Genistein	Soy	Apoptosis	[55]
	Luteolin	Fruits and vegetables	Apoptosis	[56]
	Taxol	*Taxus brevifolia*	Apoptosis	[57]
	Gallic acid	Grape seeds, rose flowers, sumac, oak, and witch hazel	Apoptosis	[58]
	Caffeic acid phenetyl ester	Propolis	Anti-proliferative	[59]
	Gingerol	Zingiber officinalis	Apoptosis	[60]
**Pancreatic**	Genistein	Soy	Anti-proliferative	[61]
	Garcinol	*Garcinia indica*	Anti-proliferative	[62]
	Limonoids	*Cipadessa baccifera*	Anti-proliferative	[63]
	Crocin	*Crocus sativus*	Apoptosis	[64]
	Fisetin	Strawberry, apple, onion, and cucumber	Apoptosis	[65]
		Pomegranate		
	Urolithin A	Fruits	Anti-proliferative	[66]
	Methyl protodioscin	Flavonoids	Anti-proliferative	[67]
	Blueberries	Flavonoids	Apoptosis	[68]
	Procyanidin		Anti-proliferative	[69]
**Colorectal**	Carotenoids	Fruits and vegetables	Anti-proliferative	[70]
	β-sitosterol	*Prunus africana*	Apoptosis	[71]
	Saponin	Soapwort plant	Apoptosis	[56]
	Genistein	Soy	Anti-proliferative	[72]
	Ellagic acid	Medicinal plants	Apoptosis	[73]
	Ferulic acid	Whole grains, spinach, parsley, grapes, rhubarb, wheat, oats, rye, and barley	Apoptosis	[74]
**Prostate**	Gallic acid	Secondary metabolite in plants	Anti-proliferative	[75]
	Neobavaisoflavone	*Psoralea corylifolia*	Apoptosis	[76]
	Rhodioflavonoside	*Rhodiola rosea*	Apoptosis	[77]
	Luteolin	Fruits and vegetables	Anti-proliferative	[78]
	Berberine	*Hydrastis canadensis*, *Berberis aristata*, *Coptis chinensis*, *Coptis japonica*, *Phellondendron amurense*, and *Phellondendron chinense Schneid*	Anti-proliferative	[79]
**Ovarian**	Corilagin	Ellagitannin in a wild of plants;	Anti-proliferative	[80]
	Gallic acid	secondary metabolite in plants;	Apoptosis	[81]
	Ellagic acid	Medicinal plants;	Anti-proliferative	[82]
	Epigallocatechin-3-gallate	Green Tea Polyphenols;	Apoptosis	[83]
	Berberine	*Hydrastis canadensis*, *Berberis aristata*, *Coptis chinensis*, *Coptis japonica*, *Phellondendron amurense* and *Phellondendron chinense Schneid*	Apoptosis/Anti-proliferative	[84]
**Blood**	Rosavin	*Rhodiola rosea*	Apoptosis	[85]
	Oleanolic acid	Fruits and vegetables	Antiangiogenic	[86]
	Silibinin	Milk thistle seeds	Antiangiogenic	[87]
	Kaempferol	Flavonoid aglycone in fruits and vegetables	Antiangiogenic	[88]
		Grape skin and seeds		
	Resveratrol	*Withania somnifera*	Antiangiogenic	[89]
	Withaferin A		Antiangiogenic	[90]

## Data Availability

Not applicable.

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
