# Peer review of "Nutraceuticals and Cancer: Potential for Natural Polyphenols"

_nutrients, 2021, doi:10.3390/nu13113834_

Round 1
Reviewer 1 Report
The present manuscript is a comprehensive review on the use of natural compounds in anticancer therapy, as ancer is one of the world’s leading causes of death.
Please can you just explain a bit more the novelty of this review and why now.
Bioavailability of each and every polyphenol differs however there is no relation between the quantity of polyphenols in food and their bioavailability in human body. Please discuss a bit more in details about the bioavailability, maybe a separate section would be recommended.
"Before absorption, these compounds must be hydrolyzed by intestinal enzymes or by colonic microflora. ....As a consequence, the forms reaching the blood and tissues are different from those present in food and it is very difficult to identify all the metabolites and to evaluate their biological activity." - Pandey KB, Rizvi SI. Plant polyphenols as dietary antioxidants in human health and disease. Oxid Med Cell Longev. 2009;2(5):270-278. doi:10.4161/oxim.2.5.9498
What about Vitamin C? Quercetin co-administration with this vitamin exerts a synergistic action.
Cancer could be prevented through healthy diets. Why only Mediterranean Diet is mentioned?
Another issue maybe to be discussed: genetics and how nutraceuticals could help?
The conclusions part needs to be improved. the conclusions have nothing to do with the nutraceuticals and cancer. Please rewrite this section highlighting the important of nutraceuticals.
Author Response
Dear Reviewer, thank you for your valuable suggestions. I hope that my responses to your comments will improve the content of the paper. In the document attached, your comments are highlighted in yellow, my answers in green.
Regards,
Jessica Maiuolo

Reviewer 2 Report
This manuscript discusses the current literature on four natural bioactive extracts, most of which are characterized by a specific polyphenolic profile. However, there are some points that should be revised:
1) End each paragraph with a two-to three-line conclusion.
2) Include figures that summarize the main findings. Please replace Figure 3, this figure should possibly depict the mode of action or general schemes of pharmacological activity.
3) The resolution of the figures is of poor quality.
4) The impact of natural bioactive extracts on the expression of microRNA, small non-coding RNAs, in cancer cells has to be reported.
5) The authors have mainly discussed the in vitro data, but they have to discuss whether these in vitro results have also been validated in in vivo models.
6) When discussing experimental studies, you should make the following general caveat: There is a possibility that most of the studies discussed in this review article were not conducted in accordance with a current consensus document that provides an outlook on best practice in pharmacological research with bioactive preparations from plants. (Reference: Izzo AA, Teixeira M, Alexander et al. A practical guide to transparent reporting of research on natural products in the British Journal of Pharmacology: Reproducibility of natural product research. Br J Pharmacol. 2020 May;177(10):2169-2178. doi: 10.1111/bph.15054. Epub 2020 Apr 16. PMID: 32298474; PMCID: PMC7174877);
7) When discussing clinical outcomes, do not overstate the clinical efficacy of the plant extracts. Rather, clearly indicate in the abstract and conclusion that further high-quality studies are needed to clearly demonstrate the clinical efficacy of the plant extracts (references: Izzo AA, Hoon-Kim S, Radhakrishnan R, Williamson EM. 2016.A Critical Approach to Evaluating Clinical Efficacy, Adverse Events and Drug Interactions of Herbal Remedies. Phytother Res. 30(5):691-700); Andrew R, Izzo AA. Principles of pharmacological research of nutraceuticals. Br J Pharmacol. 2017 Jun;174(11):1177-1194. doi: 10.1111/bph.13779. PMID: 28500635; PMCID: PMC5429327
8) More importantly, the authors should report critical comments or expert opinions on the use/effect of nutraceuticals in cancer.
Author Response

(The authors gave the same response as above.)

Round 2
Reviewer 1 Report
The authors maybe improved the manuscript, however in the latest version of available manuscript the changes cannot been seen. Please mark somehow the improvement in the manuscript.
Thank you.